# Tracking nuclear motion in single-molecule magnets using femtosecond X-ray absorption spectroscopy

Kyle Barlow [1], Ryan Phelps [1], Julien Eng [2], Tetsuo Katayama [3,4], Erica Sutcliffe [1], Marco Coletta[1], Euan K. Brechin[1], Thomas J. Penfold [2] ✉ & J. Olof Johansson [1] ✉

The development of new data storage solutions is crucial for emerging digital technologies. Recently, all-optical magnetic switching has been achieved in dielectrics, proving to be faster than traditional methods. Despite this, single-molecule magnets (SMMs), which are an important class of magnetic materials due to their nanometre size, remain underexplored for ultrafast photo-magnetic switching. Herein, we report femtosecond time-resolved K-edge X-ray absorption spectroscopy (TR-XAS) on a Mn(III)-based trinuclear SMM. Exploiting the elemental specificity of XAS, we directly track nuclear dynamics around the metal ions and show that the ultrafast dynamics upon excitation of a crystal-field transition are dominated by a magnetically active Jahn-Teller mode. Our results, supported by simulations, reveal minute bond length changes from 0.01 to 0.05 Å demonstrating the sensitivity of the method. These geometrical changes are discussed in terms of magneto-structural relationships and consequently our results illustrate the importance of TR-XAS for the emerging area of ultrafast molecular magnetism.

Magnetic materials are used to store digital information, and the magnetisation direction dictates if the storage bit is a 0 or a 1. The most important property for these materials is a bistable magnetic anisotropy that ensures the magnetisation either points up (or 0) or down (or 1) along a specific axis. Presently, magnetic data storage in servers use a small electromagnet to switch the magnetisation direction when overwriting stored data from, say, 0 to 1. Despite impressive developments associated with this method[1], read-write speeds cannot be faster than nanoseconds[2]. In the last couple of decades there has been a strong research focus on photomagnetic data recording, where femtosecond laser pulses, instead of external magnetic fields, are used to reverse the magnetisation direction within tens of picoseconds[3]. State-of-the-art methods in this area involve using either electronic[4–6] or phonon[7,8] excitation to control the magneto-crystalline anisotropy via changes in the crystal environment upon excitation. This can lead

to a torque that switches the magnetisation direction into a different orientation, which could lead to a new, faster way of storing data. The readout in this case is done using the Faraday effect[4–8].

One class of magnetic materials that have been somewhat neglected in the push toward ultrafast photomagnetic switching are single-molecule magnets (SMMs)[9]. SMMs have the added advantage of operating on the molecular scale, which could significantly increase data storage density when combined with optical techniques and plasmonic technologies to address individual molecules[10]. In Mn(III)-based SMMs, the geometric structure is closely related to the magnetic anisotropy. The high-spin $d^4$ electron configuration leads to a Jahn-Teller (JT) distortion which, with spin-orbit coupling, dictates the magnetic anisotropy[11,12]. If the metal ion exhibits an axially elongated coordination sphere, it leads to a uniaxial anisotropy along that axis. In SMMs, this uniaxial anisotropy gives rise to magnetic hysteresis below

[1]EaStCHEM School of Chemistry, University of Edinburgh, David Brewster Road, EH9 3FJ Edinburgh, UK. [2]Chemistry, School of Natural and Environmental Sciences, Newcastle University, Newcastle upon Tyne, UK. [3]Japan Synchrotron Radiation Research Institute, Kouto 1-1-1, Sayo, Hyogo 679-5198, Japan. [4]RIKEN SPring-8 Center, 1-1-1 Kouto, Sayo, Hyogo 679-5148, Japan. ✉e-mail: tom.penfold@newcastle.ac.uk; olof.johansson@ed.ac.uk

a certain blocking temperature, which is why SMMs are good candidates for data storage applications. If the distortion is axially compressed, then the complex exhibits an easy plane anisotropy in the plane perpendicular to the JT axis (Fig. 1a), meaning that the spins preferentially align along that plane. Exciting an electron from the antibonding $d_{z^2}$ to $d_{x^2-y^2}$ orbital (or vice versa) will transiently change the nature of the JT distortion and therefore the magnetic anisotropy and magnetisation direction, as shown schematically in Fig. 1a. Consequently, in a manner similar to what has been observed in metallic and dielectric systems[3], a laser pulse could be used to change the crystal-field around a metal ion and control the magnetic anisotropy. Indeed, there are reports of magnetic anisotropy switching in molecular magnets using changes in pressure[13] and temperature[14] as a driving force.

We have previously explored the photoinduced dynamics of the JT distortion in the SMM [Mn(III)$_3$O(Et-sao)$_3$(β-pic)$_3$(ClO$_4$)][15], where saoH$_2$ and β-pic are salicylaldoxime and 3-methylpyridine, respectively, now referred to as Mn$_3$, which could potentially lead to all-optical switching[16]. In this molecule, three Mn(III) ions are arranged in a triangle and the spins on each ion couple to form a $S = 6$ ground state[17]. The optical transient absorption spectroscopy (TAS) method that was used in our previous study, was limited to a convoluted picture of the electronic and nuclear dynamics because valence electronic spectra are composed of a complicated overlap between many different transitions. The important nuclear motions in the reaction coordinate were inferred from a vibronic wavepacket and quantum chemical calculations. To gain a deeper understanding of the crucial interplay between the electronic, vibrational, and spin degrees of freedom, new experimental methods are needed to probe these complicated processes. X-ray spectroscopies provide an excellent solution to this as they offer the elemental specificity and direct structural insight that optical spectroscopies cannot. In particular, time-resolved X-ray spectroscopies on femtosecond timescales carried out at X-ray free-electron lasers (XFELs), are uniquely suited to tackle such problems as they track the dynamics on and around the metal ion, exclusively.

In this paper, we present results from time-resolved K-edge X-ray absorption spectroscopy (TR-XAS)[18–23] at the SPring-8 Angstrom Compact free-electron Laser (SACLA) to gain deeper insight into the photoinduced dynamics of Mn$_3$ after metal-centred photoexcitation. With complimentary simulations, we have developed a detailed and cohesive picture of the nuclear dynamics of Mn$_3$ after photoexcitation. We show that most aspects of the TR-XAS measurement can be interpreted considering motion along a single Jahn-Teller mode with a frequency of 181 cm$^{-1}$. The changes in the main and pre-edge regions can be simulated with good agreement to experimental data with bond length changes in the inner coordination sphere up to 0.05 Å. Using the time dependence of the dynamics at the main edge, we can accurately fit time constants that match those found in optical TAS data and provide evidence of coherent motion at a frequency of around 180 cm$^{-1}$, which further supports its assignment as the dominant mode in the reaction coordinate. Measurements focussing on the pre-edge region, which provides sensitivity to the coordination symmetry, agree well with the changes in the inner coordination sphere suggested from the main edge. The elemental specificity of TR-XAS in combination with high-level computations has allowed tracking the crystal-field dynamics in large polynuclear complexes with sensitivity to bond length changes on the order of 0.01 Å. This level of detail will be important in the emerging field of ultrafast photoswitchable SMMs.

## Results and discussion

A basic description of the experiment is shown in Fig. 1. The results are presented in Fig. 2a, which shows the experimental (red) and calculated (blue) ground state K-edge X-ray absorption spectrum of Mn$_3$. The experimental and calculated spectra are in good agreement, and show a weak pre-edge (1$s$ to 3$d$ transitions) at 6539 eV and the main edge at 6549 eV, matching what is expected from Mn(III) ions[24,25]. The absorption difference spectrum at 700 fs, obtained after photoexcitation of Mn$_3$ at 400 nm, is shown in Fig. 2b. This excitation wavelength corresponds to the highest energy crystal-field transition, but previous studies indicate that the lowest excited ligand-field state is populated within 100 fs (as shown in Fig. 1a)[15]. At this pump-probe delay, there is an increase in the pre-edge intensity and a small redshift of the main edge of around 0.2 eV (Supplementary Fig. 1) at 700 fs.

To extract kinetic information, the largest pump-induced transient signal at 6548.5 eV was measured as a function of time delay. The decay signal could be fit with a biexponential function, as shown in Fig. 2c. Interestingly, there were oscillations superimposed on the transient signal. This type of behaviour is indicative of coherent vibrational motion induced by the pump pulse and was also observed in our optical TAS data, due to the $d_{z^2}$ to $d_{x^2-y^2}$ switch[15]. We therefore

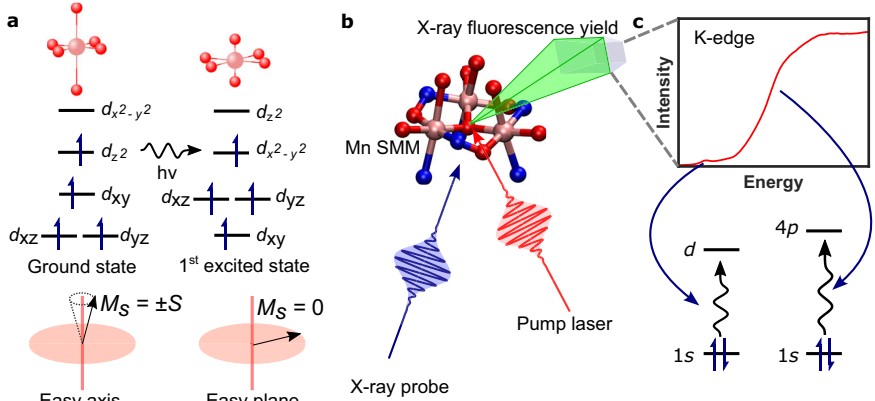

**Fig. 1 | Overview of the pump-induced dynamics and the information the X-ray absorption probe can provide. a** Excitation of an electron from the $d_{z^2}$ to $d_{x^2-y^2}$ changes the JT distortion from axial elongation to axial compression. Bottom: in the electronic ground state, the magneto-crystalline anisotropy is easy axis type where the spin magnetic moment aligns along the anisotropy axis. Therefore, the lowest energy configuration occurs when the $z$-component of the spin quantum number $M_s = \pm S$. After excitation and the JT switch, the complex exhibits easy plane anisotropy and the spin magnetic moment preferentially aligns in the plane perpendicular to the anisotropy axis with the lowest energy state $M_s = 0$. **b** Basic schematic of the experiment. A pump laser excites the sample (Mn$_3$) and an intense X-ray beam is used to measure the K-edge spectra of the sample before and after excitation using total fluorescence detection. The intensity of the spectra is given by the total fluorescence yield. The peripheral atoms of the Mn$_3$ have been removed for clarity. Mn, pink; O, red; N, blue. **c** K-edge spectrum and transitions associated with the spectral features.

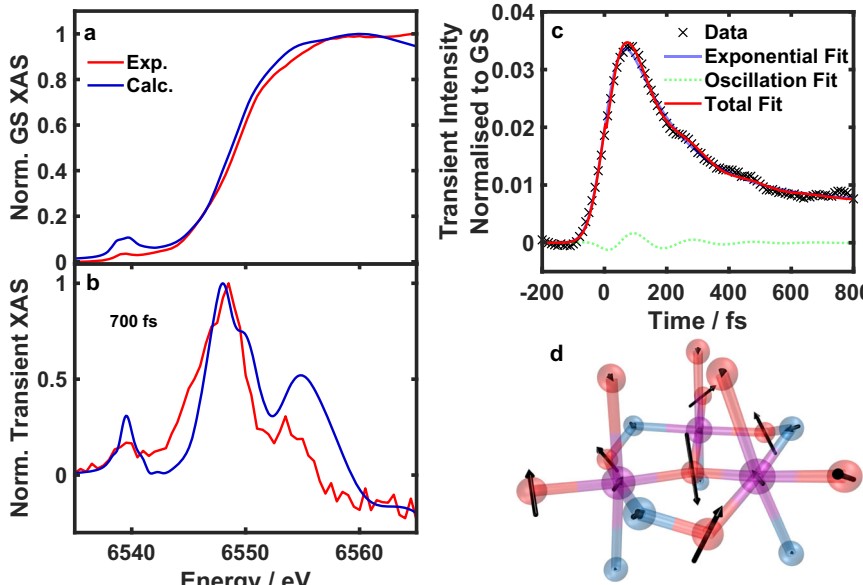

**Fig. 2 | X-ray K-edge absorption spectra of Mn₃. a** The experimental (red) and calculated (blue) ground state K-edge XAS of Mn₃. **b** XAS difference spectra at 700 fs after 400 nm photoexcitation. The calculated spectrum was simulated using the 74% excitation yield and calculating the XAS predominantly along *v*60 to give the best match to the experimental results. Further details concerning the computations can be found in the methods section. **c** Time evolution of the TR-XAS signal at an X-ray probe energy of 6548.5 eV. The fitted line is made up of two exponential decay components with time constants $\tau_1 = 178 \pm 6$ fs and $\tau_2 = 9000$ fs.

There is also a small oscillatory component with a frequency of $172 \pm 8$ cm⁻¹ that is a signature of vibrational coherence. The data have been smoothed with a five-point adjacent average (the effect of smoothing is shown in Supplementary Fig. 2). The signal is normalised with respect to the ground state (laser off) signal. **d** Schematic of the mode (*v*60) that is responsible for to oscillation seen in **c**, where the peripheral ligands of the Mn₃ complex have been removed for clarity. Mn, purple; O, red; N, blue. A video of this mode is provided in Supplementary Movie 1. Source data are provided in the Source Data folder.

included a term in the fitted equation to capture the vibrational wavepacket:

$$\Delta I(t) = \mathrm{IRF}(\sigma, t) * \left( \sum_{i=1}^{2} A_i e^{-\frac{t}{\tau_i}} + A_{\mathrm{osc}} e^{-\frac{t}{\tau_{\mathrm{osc}}}} \cos(2\pi c \tilde{\nu} t) \right) \quad (1)$$

where $A_i$ and $\tau_i$ are the exponential decay amplitudes and decay times, respectively. $A_{\mathrm{osc}}$ and $\tau_{\mathrm{osc}}$ are the wavepacket amplitude and dephasing time, respectively. The wavepacket frequency is given by $\tilde{\nu}$, and $c$ is the speed of light. The equation is convoluted with an instrument response function (IRF) modelled as a Gaussian of width $\sigma = 42 \pm 2$ fs from the fitting, which takes into account the finite width of the pump and probe pulses. The fitting yielded a decay constant of $\tau_1 = 178 \pm 6$ fs in agreement with the optical data[15] and the excited state lifetime was fixed at $\tau_2 = 9000$ fs[15]. The value of the oscillation frequency was $\tilde{\nu} = 172 \pm 8$ cm⁻¹. This value is close to the frequency of the calculated normal mode *v*60 at 210 cm⁻¹, which involves an in-phase oscillation of the three Mn(III) ions along their JT axes (Fig. 2d and Supplementary Movie 1) and was also observed in the optical transient absorption with a frequency of $\tilde{\nu} = 181 \pm 3$ cm⁻¹. The vibrational dephasing time was found to be shorter than in the optical data at $\tau_{\mathrm{osc}} = 170 \pm 80$ fs. This is likely due to the lower amplitude of the oscillations in comparison to the optical data which means the oscillations fall below the noise level at earlier pump-probe delay times compared to the optical TAS data.

The main edge is particularly sensitive to the structure of the inner coordination sphere and studies have shown it is possible to extract bond length changes in the excited state using TR-XAS[21]. We have previously proposed that the structural rigidity of Mn₃ constrains the reaction coordinate to a single dominant mode, *v*60, in contrast to the more flexible monomer Mn(acac)₃[15]. Considering the agreement in the optical and X-ray data, we aimed to interpret the TR-XAS data within the normal mode framework by simulating XAS spectra of Mn₃ at different geometries along mode *v*60 in the electronic ground state

to extract structural parameters. We could not calculate this in the excited state due to the difficulty in modelling exchanged-coupled polynuclear complexes with large spin angular momenta. It is expected that the electronic excitation of metal-centred transitions should lead to only small modulations of the overall X-ray spectra as there is no formal oxidation/reduction of the metal centres. The simulated difference spectrum shown in Fig. 2b was generated by calculating the XAS spectrum at the ground state geometry and at a different geometry along *v*60 corresponding to very small geometry changes. The proportion of excited molecules (74%, see Supplementary Information) was taken into account to compare with experiment. Considering the complexity of the molecule, there is an excellent agreement with experiment showing the peaks at the pre-edge, rising edge and post-edge and also the negative signal above 6560 eV. This strongly suggests that the TR-XAS data can be accurately described by structural changes along the dominant *v*60 mode. The changes in nuclear coordinates are detailed in Supplementary Table 1. In particular, the axial Mn−O bond lengths increase by around 0.05 Å and the Mn−N axial bond lengths decrease by 0.03 Å suggesting the manganese ions shift towards the β-pic ligand along the JT axis. Equatorial bonds only change by ≤ 0.011 Å indicating restriction of in-plane motion as previously suggested[15]. The effect of other modes alongside *v*60 is shown in Supplementary Fig. 3 and the eigenvectors of these modes are shown in Supplementary Fig. 4. While the combination of multiple modes can improve the agreement with the experimental difference spectrum, these additional modes only give rise small modulations of the spectra.

In order to explore the structural changes in more detail, we carried out energy scans at different time delays at the pre-edge spectral region. Pre-edge spectra are particularly sensitive to coordination sphere symmetry and the ligand-field, and therefore are an excellent marker of electronic and nuclear crystal-field dynamics. Figure 3a displays pre-edge spectra, with the background from the tail

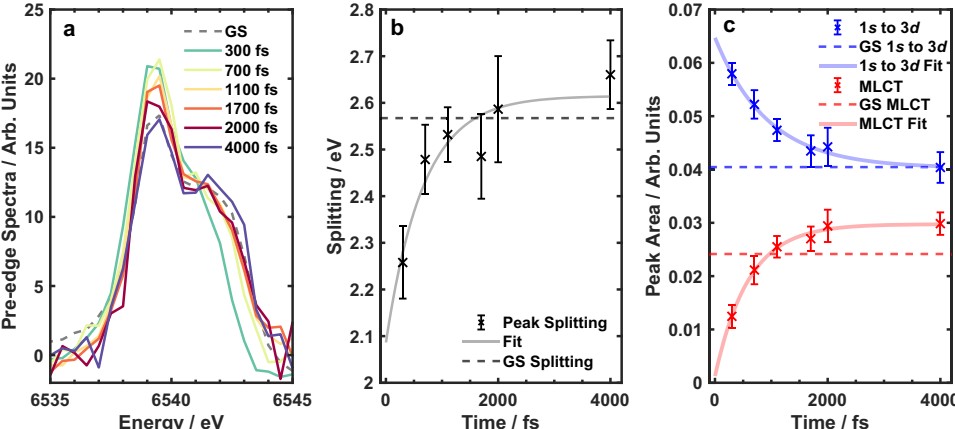

**Fig. 3 | Pre-edge dynamics after photoexcitation. a** Ground state (GS) and excited state pre-edge after subtracting background of main edge. There is a clear increase in the intensity of the low energy 1s to 3d band (6539.4 eV) after photoexcitation that then decays. There is a redshift of the high energy MLCT peak (6542 eV) initially that then blueshifts back to the GS value. **b** Energy splitting of the two peaks as a function of time calculated using Gaussian fits (Supplementary Fig. 9). The fitted line is an exponential decay with a time constant of 700 fs. **c** Peak areas of the two pre-edge peaks as a function of time calculated using Gaussian fits (Supplementary Fig. 9). The fitted lines are an exponential decay with time constants of 900 fs for the 1s to 3d peak and 600 fs for the MLCT peak. The dashed horizontal lines show the calculated values for the GS pre-edge. The error bars in panels **b** and **c** are calculated from the uncertainties associated with the least squares fitting presented in Supplementary Fig. 9 and describe a 95% confidence interval. Source data are provided in the Source Data folder.

of the main edge subtracted (details shown in Supplementary Figs. 5 and 6) of the ground state and excited state at different pump-probe delay times. The spectra are comprised of a low energy 1s to 3d peak at 6539.4 eV and a high energy peak at around 6542 eV[25]. According to the time-dependent density functional theory (TDDFT) calculations (see Methods), the higher energy transition is of 1s to orbitals of mixed 3d and ligand character, in agreement with previous literature regarding manganese K-edge XAS[25]. Considering the metal-ligand charge-transfer character of this transition, we use the terminology in the field and refer to it as the MLCT peak. The data show an initial increase in the 1s to 3d transition intensity after photoexcitation which then decays and returns close to the ground state value. Immediately after excitation, the MLCT peak is redshifted compared to the ground state but then blueshifts over the next few picoseconds. We see qualitative agreement with the simulated spectra at different geometries along mode v60 (Supplementary Fig. 7). There is a decrease in intensity of the 1s to 3d peak and a blueshift of the MLCT peak moving from the distorted structure back to the ground state. Kinetic scans at three probe wavelengths were also carried out and are shown in Supplementary Fig. 8. These can be fit to the same time constants as the main edge and prove there is consistency across the whole K-edge spectra.

To quantify these changes, the spectra were fit with two Gaussian functions (fits shown in Supplementary Figs. 5 and 9) with the peak centre of the 1s to 3d transition fixed at 6539.4 eV. The time evolution of the fitting parameters is shown in Figs. 3b and 3c along with the ground state values. Figure 3b shows the peak areas as a function of time. The 1s to 3d peak decays at the same rate the MLCT peak grows, showing they are likely sensitive to the same dynamical process. Considering the pump pulse only excites a ligand-field transition, the total number of d-electrons is constant and the growth in the 1s to 3d peak is likely due to a change in the coordination symmetry. Returning to the structural parameters obtained from the simulations, there are very small changes in the equatorial plane (Supplementary Table1) due to the rigid in-plane bonds. The axial bonds in the ground state are about equal at 2.37 Å but in the excited state the Mn-O bonds increase by about 0.05 Å and the Mn-N bonds decrease by around 0.03 Å. This suggests a reduction in symmetry along the z-axis which will cause the 1s to 3d peak to increase in intensity due to 3d-4p mixing in good agreement with the experimental peak area. In addition, there are very small bond angle changes (≈ 1°) in the inner coordination sphere (Supplementary Table 1) that could also contribute to the lowering of symmetry. Upon photoexcitation, there is no change in the manganese ion oxidation state and hence no change in the effective nuclear charge. Therefore, the 3d atomic orbital energy will remain nearly constant and the 1s to 3d pre-edge peak will show no significant shifts. The simulated spectra (Supplementary Fig. 7) show no change in the 1s to 3d peak maximum, which is in agreement with the experimental data.

The peak centre and area of the MLCT transition also change with time. Upon photoexcitation, all metal-ligand bonds elongate apart from the β-pic ligand bond (Supplementary Table 1). Compared to the Franck-Condon structure, there is less overlap between the ligand and metal 3d orbitals in the excited state distorted structure which leads to the MLCT transitions having more 1s to 3d character than 1s to ligand character. This has two effects: the peak centre redshifts as the 3d orbitals are at a lower energy than the ligand orbitals and the intensity of the transition will decrease as the 1s to 3d transition is more forbidden than the 1s to ligand transition. Both these effects are observed in the experimental data at early times, but subsequently the peak centre and area then return to near their ground state values over the next few picoseconds. Similar dynamics are found in the 1s to 3d and MLCT peak as changes arise from the same dynamics associated with the inner coordination sphere.

In the context of photomagnetism, there are generally two important structural parameters that will dictate the magnetic properties in $Mn_3$[17]. In the ground state, the magnetic property is dominated by ferromagnetic superexchange between the three Mn(III) sites, yielding a total spin of $S = 6$. The sign of the superexchange interaction is dictated mostly by the dihedral Mn−N−O−Mn angles. From the ground state to the distorted structure, these decrease by around 0.5°, from 43 to 42.5°. Antiferromagnetically coupled Mn(III) triangles generally have dihedral angles smaller than 20°[17], therefore, it is very unlikely that photoexcitation changes the sign of superexchange interaction. The second parameter is the Mn-ligand bond lengths, particularly the difference in the equatorial and axial bond lengths as these determine the magnetic anisotropy through their influence on the crystal field. The average equatorial bond length only increases by around 0.01 Å from the Franck-Condon to the distorted structure, whilst the axial bonds change by around 0.05 Å. The metal-ligand bond

lengths in Mn(acac)$_3$ are calculated to change by around 0.2 Å[15], which is an order of magnitude more than Mn$_3$. This is further evidence that this molecule is very rigid and is likely the reason for the short 10 ps excited state lifetime in comparison to the much more flexible Mn(acac)$_3$, which has a lifetime well over 1 ns[15]. In light of these small structural changes, it is unlikely that changes in nuclear structure alone will lead to methods of magnetisation control in such a rigid poly-nuclear complex. If control of magnetisation is to be achieved in these large metallic Mn(III) clusters by influencing the geometry of the crystal field, perhaps more focus should be given to flexible molecules that will be able to accommodate the larger changes in nuclear coordinates. There is some evidence that light can influence the magnetic properties in some Mn$_{12}$ SMMs[26], and so with the right molecular structure fs magnetic manipulation might be possible. We have previously shown in Mn(III) monomer model systems, i.e. not SMMs, that it is possible to simplify the reaction coordinate to a reduced number of normal modes by introducing a static rather than dynamic JT axis[27]. Also, excited state lifetimes can be reduced by adding structural rigidity into the equatorial plane, which restricts the in-plane expansion and favours ground state recovery as suggested here[28]. Therefore, it could be possible to direct nuclear motion in the excited state using synthetic design towards the goal of ultrafast magnetisation control.

Considering the discussion of the TR-XAS results presented in the previous paragraphs, all of the data can be explained by motion along one reaction coordinate dominated by mode $v60$. This may be a general finding for molecules where there are only small transient XAS signals due to the excitation of metal-centred transitions that do not change oxidation state or spin state but can cause geometrical distortions. To the best of our knowledge, all previous work using femtosecond K-edge TR-XAS studied metal complexes after generation of a charge-transfer state, where there is a formal change in metal oxidation state and therefore large transient signals due to large shifts in the main edge. The measurements presented here show that even upon excitation of metal-centred transitions in large polynuclear exchange-coupled complexes, conclusions can be drawn from comparison to computation with sensitivity to bond-length changes on the order of hundredths of ångströms.

The modern information technology *era* has led to the ever-increasing importance of data storage, and consequently to advance this field it is critical that researchers develop a deeper understanding of the electronic, nuclear and spin dynamics and their coupling within molecular materials as a way to develop these technologies. In this work, we have demonstrated that time-resolved K-edge X-ray absorption spectroscopy is an excellent method for tracking the nuclear dynamics even in complex polynuclear transition metal complexes. The whole K-edge difference spectra, after metal-centred excitation, can be interpreted using geometric changes occurring predominantly along a single JT active mode, which is crucial for the magnetic anisotropy in these systems. The signature of this mode was also observed as an oscillation with a frequency of around 180 cm$^{-1}$ in the kinetic trace, suggesting a coherent excitation of this mode, in excellent agreement with the optical time-resolved measurements[15]. Spectral changes in the pre-edge also agree well with the changes across the whole K-edge region with motion along one JT mode being dominant. The good agreement between computation and experiment allowed us to estimate bond length changes from the Franck-Condon structure to the relaxed excited state on the order of hundredths of ångströms demonstrating the sensitivity of TR-XAS. Changes in the axial JT bonds were greater than 0.03 Å whereas equatorial bonds elongated only by around 0.01 Å indicating structural rigidity in the equatorial plane. In light of these small geometry changes upon excitation, we suggest examining at more flexible SMMs as a platform to direct nuclear motion using light in order to control the magnetisation direction in SMMs. Although we have closely studied the nuclear structure in this paper, electronic and spin dynamics will certainly play a role in magnetisation dynamics in SMMs. Studying these dynamics with methods that are more sensitive to *d*-orbital occupation such as resonant inelastic X-ray scattering[29], or spin sensitive such as time-domain terahertz spectroscopies – both pump[30] and probe[31] – will be vital in the development of nanoscale magnetic data storage devices.

## Methods

### Experimental details

Mn$_3$ was synthesised according to literature procedures[17]. A 9 mM ethanol solution of Mn$_3$ was made and this was injected into the sample chamber[32] using a glass nozzle that produced a 50 μm cylindrical jet at a speed of 12 ms$^{-1}$.

The TR-XAS measurements were performed at BL3[33] of SACLA[34]. The X-ray absorption was measured using the total fluorescence detection method. For both incident and total fluorescence X-ray intensity measurements, we used Si photodiodes (Hamamatsu, S3590–09). The geometry of these detectors is described in Ref. 32. Two Si(111) crystal/s in the (+,−,−,+) geometry were used to monochromate the beam and a Be compound refractive lens was used to focus the X-rays at the sample position to a size of 10 (H) x 10 (V) μm$^2$ at FWHM. At the interaction point, 400 nm laser pulses were focussed down to 210 (H) x 215 (V) μm$^2$ at FWHM and interleaved with the X-ray pulses to measure both optical laser on and off spectra to acquire the pump induced difference spectra. The timing jitter was measured using the beamline diagnostics[35] and corrected in the data analysis, to improve the time resolution. We estimate that X-ray flux per pulse was ca. 4 μJ. This estimation is based on the bandwidth of the Si double crystal monochromator (which only preserves ca. 2% of the SASE pink beam energy of 600 μJ) and that transmission through the Be compound refractive lens is 40% at 6.5 keV, leaving ca. 4 μJ/pulse.

A power titration was carried out to ensure single photon excitation and to minimize nonlinear processes, the results of this are shown in Supplementary Fig. 7. A pump fluence of 13.3 mJcm$^{-2}$ was used corresponding to a 74% excitation yield.

### Calculations

Geometry optimisation and normal mode calculations of the Mn$_3$ complex were performed using the density functional theory (DFT) with the PBE functional[36] using a DKH-def2-SVP basis set[37] including the relativistic effects via the 2nd order Douglas-Kroll-Hess formalism[38–40]. The calculations were performed with the Orca 4 quantum chemistry package[41]. Simulations of the Mn K-edge XANES spectra were performed using the FDMNES package[42,43]. Throughout a self-consistent muffin-tin-type potential of radius 6.0 Å around the absorbing site was employed. The interaction with the X-ray field was described using the electric quadrupole approximation, and scalar relativistic effects were included. To transform the computed cross-sections into XANES spectra that can be compared to experiment, the cross-sections were convoluted with a function that accounts for the core-hole-lifetime broadening, instrument response, and many-body effects, e.g., inelastic losses. Throughout this work, this convolution has been performed using an energy-dependent arctangent function[44] via an empirical model close to the Seah-Dench formalism[45].

XAS were computed as a function of displacements along dimensionless normal coordinates (DNC). In practice, we have generated a set of geometries that each correspond to a displacement along a given DNC. The displacement along mode $v60$ was chosen by simple comparison of the calculated XAS of distorted geometries to the experimental ones. Because experimentally, not all molecules are excited, the resulting XAS will include both molecules in the excited states and in the ground states. In order to take into account this effect in the calculation, the ratio of the ground state to excited state signal must be simulated, which will modulate the amplitude of the signal.

Pre-edge simulations were performed using Restricted Excitation Window Time-dependent Density Functional Theory (REW-TDDFT)[46]

within the TPSSh[47] exchange and correlation functional and DKH-def2-SVP basis set[37] as implemented in the ORCA quantum chemistry package[41]. Scalar relativistic effects were included using the 2nd order Douglas Kroll Hess method. All simulations included 15 excited states where the interaction with the X-ray field was described using the electric quadrupole approximation[48].

Structural distortions to interpret the experimental transient spectra Mn K-edge spectra were performed along the lowest 100 normal modes and generated using the VCMaker package[49,50]. Normal modes consistent with the experimental observations were then contained to generate the excited state geometry.

## Data availability
Raw data were generated at SACLA and are available from the corresponding authors. Derived data used to create the figures shown in the paper are provided in "Source Data.zip" attached to this paper. Source data are provided with this paper.

## Code availability
VCMaker used to generate the normal mode distortions can be found at https://github.com/JulienEng/VCMaker/.

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

## Acknowledgements

JOJ, EKB, and TJP acknowledge funding from the EPSRC (EP/V010573/1, EP/W008009/1, EP/X035514/1 and EP/X026973/1). T.K. acknowledges JSPS KAKENHI for Grants JP19H05782, JP21H04974, and JP21K18944. The experiment was performed with the approval of the Japan Synchrotron Radiation Research Institute (JASRI; Proposal No. 2021B8001).

## Author contributions

KB synthesised and characterised the sample under guidance from MC and EKB. TK performed the experiments at SACLA, with remote, online participation by KB, RP, ES, and JOJ. KB and RP processed and analysed the data. JE and TJP performed all calculations. KB, JE, TJP and JOJ discussed and interpreted the results, and wrote the paper, with contributions from all authors.

## Competing interests

The authors declare no competing interests.
