## [Peer Review File · Nature Communications]

Tracking nuclear motion in single-molecule magnets using femtosecond X-ray absorption spectroscopyReviewer #1 (Remarks to the Author):

The authors present an interesting study of a single-molecule magnet, where the spin configuration of three manganese atoms leads to a magnetic configuration. With an optical pump pulse, a d-electron is promoted to an excited state at the metal atoms, which in turn changes the symmetry of the d-shell configuration and triggers a rearrangement of the nuclei around the metal atoms. With time-resolved X-ray absorption spectroscopy at the Mn K-edge, the authors study the geometrical and electronic changes by analysing the pre-edge and main line features of the spectra. They find changes that they interpret with the excitation of a single main vibrational normal mode, that they observe to be strongly damped though with still visible coherent oscillations. The excited state decays on the timescale of several picoseconds back to the ground state.

The authors motivate the choice of the sample with potential future applications for magnetic storage. Their results though show that the chosen molecule does not have the necessary properties: The changes in the nuclear configuration are too small for being useful as magnetic storage. The authors interpret this with the apparent stiffness of the molecule as a result of three coupled Mn atoms. The authors speculate that this insight provides a recipe towards magnetic applications of single-molecule magnets with fast switching properties.

The presented work is significant in the field of molecular dynamics studies in metal-coordinated molecules, but also shows insight into ultrafast magnetization dynamics. The presented results and interpretations thus bridge ideas from molecular physics towards solid-state physics and applied physics fields.

The study of the authors is carefully designed with interesting supporting optical studies (previously published), a careful power titration and a detailed analysis of the X-ray absorption data. The analysis is sound, all underlying details are presented (at least in the supporting material) and the conclusions are carefully balanced and well-based in the experimental data. The work complies with all high standards in the field and provides enough detail and references for a reproduction of the results.

I have no further comments to the paper and recommend publication without further review. Some small typos / wrong words though should be fixed in the editorial phase.

Reviewer #2 (Remarks to the Author):

The authors discuss a femtosecond X-ray absorption study on a Trinuclear Mn based molecular "magnet"

1. First things first: is the system a "magnet" ? It S=6 configuration makes the molecular sensitive to magnetic field but I don't think that there is any coercive field that I would see as essential ingredient of being a magnet. If this is not "required", I would suggest the authors to comment to clarify for the readers less used to the terminology

Going to the content of the body of the work, I think it is a quite interesting work, with good (but not exceptional) signal to noise and the analysis is sound .. but I have a bit of a difficult time seeing this work of interest for the wider community.

I divide my comments in two classes (x-ray and system)

* X-ray *

2. As a "x-ray spectroscopy" paper, it does bring something somewhat new: the pre-edge analysis. The authors state that this peak is of MLCT origin but I am confused by this name. Usually, in transition metal compounds papers, MLCT bands are absorption bands (1-2 eV above the ground state) where the electron can be transferred from one of the lower lying 3d orbitals. Are the authors suggesting that the peak at 6452 this is a 1s -> ligand+3d transition ?

If so this should be clarified a bit more to avoid confusion.

Also the reason why this band should change more than the first so called "3d peak" is not very clear to me.

Despite being an interesting discussion, I think that the pre-peak evolution does not bring fundamentally new information. A decrease of JT distortion is not very surprising at higher energies.

3. the authors should clarify how the XAS signal was calculated. My understanding is that structural changes along some specific modes (figure SI3) were used to generate new structures. How the amplitude of the displacement was chosen is not clear to me nor it is clear how these amplitude are coupled to the "assumed excitation fraction". Also parameters used for FDMNES should be mentioned.

* system *

4. It is not clear how the specific system could be used in a all-optical device. How would you read and write ? This is connected to the point 1. In normal magnet a stronger than coercive field is used to change the magnetization... Once changes the magnetization is stable for very long time. What about the Mn₃ system ?

* minor point *

more specific information on the experimental setup should be provided, flux per X-ray pulse, kinds of detectors used. One can only guess that the signal was measured in total fluorescence yield ...

All is all I feel that this paper tries to struck a balance between the new opportunities in time-resolved XAS and the interest in molecular magnet. In practice though, I feel that both aspect are not quite compelling enough for publication in Nature Communication. The X-ray data are nice but they are not striking enough to be a landmark in the field, the system itself, despite of obvious interest for photo-physicists, is not quite the best magnet due to the short lifetime at the very least.

Reviewer 1:

The authors present an interesting study of a single-molecule magnet, where the spin configuration of three manganese atoms leads to a magnetic configuration. With an optical pump pulse, a d-electron is promoted to an excited state at the metal atoms, which in turn changes the symmetry of the d-shell configuration and triggers a rearrangement of the nuclei around the metal atoms. With time-resolved X-ray absorption spectroscopy at the Mn K-edge, the authors study the geometrical and electronic changes by analysing the pre-edge and main line features of the spectra. They find changes that they interpret with the excitation of a single main vibrational normal mode, that they observe to be strongly damped though with still visible coherent oscillations. The excited state decays on the timescale of several picoseconds back to the ground state.

The authors motivate the choice of the sample with potential future applications for magnetic storage. Their results though show that the chosen molecule does not have the necessary properties: The changes in the nuclear configuration are too small for being useful as magnetic storage. The authors interpret this with the apparent stiffness of the molecule as a result of three coupled Mn atoms. The authors speculate that this insight provides a recipe towards magnetic applications of single-molecule magnets with fast switching properties.

The presented work is significant in the field of molecular dynamics studies in metal-coordinated molecules, but also shows insight into ultrafast magnetization dynamics. The presented results and interpretations thus bridge ideas from molecular physics towards solid-state physics and applied physics fields.

The study of the authors is carefully designed with interesting supporting optical studies (previously published), a careful power titration and a detailed analysis of the X-ray absorption data. The analysis is sound, all underlying details are presented (at least in the supporting material) and the conclusions are carefully balanced and well-based in the experimental data. The work complies with all high standards in the field and provides enough detail and references for a reproduction of the results.

I have no further comments to the paper and recommend publication without further review. Some small typos / wrong words though should be fixed in the editorial phase.

ANSWER: We thank the reviewer for their kind and supporting comments. We have gone through the manuscript again and have corrected any typos/wrong words.

Reviewer 2:

The authors discuss a femtosecond X-ray absorption study on a Trinuclear Mn based molecular "magnet"

1. First things first: is the system a "magnet" ? It $S=6$ configuration makes the molecular sensitive to magnetic field but I don't think that there is any coercive field that I would see as essential ingredient of being a magnet. If this is not "required", I would suggest the authors to comment to clarify for the readers less used to the terminology

ANSWER: Yes, it is. In this family of $[Mn_3]$ complexes ferromagnetic nearest neighbour (Mn^{III} , $s = 2$) exchange leads to a ground state spin of $S = 6$. This is split by an axial zero field splitting parameter in the range $D = -0.4$ to -0.9 cm^{-1} (originating from the single ion anisotropy of the Mn^{III} ions), such that there is an energy barrier to the reversal of the spin from $M_s = +6$ to $M_s = -6$ in the range $U_{eff} = \sim 25-57$ K. This results in the observation of slow relaxation of the magnetization, as manifested by both out-of-phase ac susceptibility signals and temperature and sweep rate dependent hysteresis loops in magnetization versus field studies. Because of the similarity to bulk magnets, these species have been referred to as single-molecule magnets (SMMs). This is described in detail in reference 9. Further details specific to the $[Mn_3]$ class of molecules is provided in reference 17. We have added the following clarification based on the reviewer's suggestion:

One class of magnetic materials that have been somewhat neglected in the push toward ultrafast photomagnetic switching are single-molecule magnets (SMMs).⁹ SMMs have the added advantage of operating on the molecular scale, which could significantly increase data storage density when combined with optical techniques and plasmonic technologies to address individual molecules.¹⁰ In Mn(III)-based SMMs, the geometric structure is closely related to the magnetic anisotropy. The high-spin d^4 electron configuration leads to a Jahn-Teller (JT) distortion which, with spin-orbit coupling, dictates the magnetic anisotropy.^{11,12} If the metal ion exhibits an axially elongated coordination sphere, it leads to a uniaxial anisotropy along that axis. In SMMs, this uniaxial anisotropy gives rise to magnetic hysteresis below a certain blocking temperature, which is why SMMs are good candidates for data storage applications.

Going to the content of the body of the work, I think it is a quite interesting work, with good (but not exceptional) signal to noise and the analysis is sound .. but I have a bit of a difficult time seeing this work of interest for the wider community.

ANSWER: We believe it will be of interest to researchers in molecular magnetism and inorganic photophysics because measuring and manipulating spins in inorganic complexes by light is a very active topic. At the same time, it will also stimulate the large research community in condensed matter physics working on ultrafast magnetism. In all these areas, there are large research communities active in the US, Germany, France, Spain, and Japan among other. We understand that the signal-to-noise might not be exceptional (but good as the reviewer says), however, the data are exceptional as we have been able to determine bond length changes as small as 0.01 Å, which we really think is outstanding and shows the power of the method. Therefore, we also believe that the results will be of interest to the whole photophysics and XFEL communities. Based on the large numbers of researchers in all of these fields, and the wide span within both chemistry and physics, and even potentially technology, we believe that the results will indeed be of interest to the wider community.

I divide my comments in two classes (x-ray and system)

* X-ray *

2. As a "x-ray spectroscopy" paper, it does bring something somewhat new: the pre-edge analysis. The authors state that this peak is of MLCT origin but I am confused by this name. Usually, in transition metal compounds papers, MLCT bands are absorption bands (1-2 eV above the ground state) where the electron can be transferred from one of the lower lying 3d orbitals. Are the authors suggesting that the peak at 6452 this is a 1s -> ligand+3d transition ?

If so this should be clarified a bit more to avoid confusion.

ANSWER: We fully understand the confusion about nomenclature but researchers in the XAS field call transitions such as the 1s -> ligand+3d transition MLCT, which is why we have used this name. We have changed the manuscript to make this clearer:

The spectra are comprised of a low energy 1s to 3d peak at 6539.4 eV and a high energy peak at around 6542 eV. According to the TDDFT calculations (see Methods), the higher energy transition is of 1s to orbitals of mixed 3d and ligand character, in agreement with previous literature regarding manganese K-edge XAS.²⁵ Considering the metal-ligand charge-transfer character of this transition, we use the terminology in the field and refer to it as the MLCT peak.

Also the reason why this band should change more than the first so called "3d peak" is not very clear to me.

ANSWER: This band is expected to change more than the 3d peak because it involves orbitals that are more delocalised and therefore will be more affected by the nuclear motions in the excited states. In comparison the 1s->3d excitation is very local and will be much less affected by the motions in the excited state. Furthermore, considering that the initial photoexcitation does not change the oxidation state of the Mn ions, there is likely only a small change in the effective nuclear charge and

therefore no significant change in the 3d orbital energies. The manuscript text has been amended to:

In addition, there are very small bond angle changes ($\approx 1^\circ$) in the inner coordination sphere (Table S11) that could also contribute to the lowering of symmetry. Upon photoexcitation, there is no change in the manganese ion oxidation state and hence no change in the effective nuclear charge. Therefore, the 3d atomic orbital energy will remain nearly constant and the 1s to 3d pre-edge peak will show no significant shifts. The simulated spectra (Figure S17) show no change in the 1s to 3d peak maximum, which is in agreement with the experimental data.

Despite being an interesting discussion, I think that the pre-peak evolution does not bring fundamentally new information. A decrease of JT distortion is not very surprising at higher energies.

ANSWER: We are encouraged that the reviewer agrees that the pre-edge analysis brings something new to the field (*“As a x-ray spectroscopy paper, it does bring something somewhat new: the pre-edge analysis”*), but accept that it might not be contributing to new fundamental knowledge of pre-edge spectra in general. We argue, however, that the pre-edge analysis forms an important part of the whole interpretation, which confirms our conclusions that the molecule is transiently changing the Jahn-Teller distortion in the excited state. This is very important information given that the key to all-optical magnetic switching is in controlling this distortion/anisotropy. These XFEL measurements show for the first time very conclusive evidence that this takes place, and it is impressive that we can see this in such a large metal complex with three metal ions – this is indeed the novelty of our work.

3. the authors should clarify how the XAS signal was calculated. My understanding is that structural changes along some specific modes (figure S13) were used to generate new structures. How the amplitude of the displacement was chosen is not clear to me nor it is clear how these amplitude are coupled to the "assumed excitation fraction". Also parameters used for FDMNES should be mentioned.

ANSWER: XAS were computed as a function of displacements along dimensionless normal coordinates (DNC). In practice, we have generated a set of geometries that each correspond to a displacement along a given DNC. The displacement along mode 60 was chosen by simple comparison of the calculated XAS of distorted geometries to the experimental ones. Because experimentally, not all molecules are excited, the resulting XAS will include both molecules in the excited states and in the ground states. In order to take into account this effect in the calculation, the ratio of the ground state to excited state signal must be simulated, which will modulate the amplitude of the signal. Each Mn K-edge spectrum was simulated using the multiple scattering theory method as implemented within the FDMNES package (Joly, Y. X-ray Absorption Near-Edge Structure Calculations Beyond the Muffin-Tin Approximation Phys. Rev. B 2001, 63, 1251201– 12512010) using a self-consistent potential of radius 6.0 Å around the absorbing atom. The interaction with the X-ray field was described using the electric quadrupole approximation. Following the calculation, many body effects and the core hole lifetime broadening were accounted for using an arctangent convolution (Bunău, O.; Joly, Y. Self-Consistent Aspects of X-ray Absorption Calculations J. Phys.: Condens. Matter 2009, 21, 345501).

We have added the following text to the manuscript:

Geometry optimisation and normal mode calculations of the Mn_3 complex were performed using the density functional theory (DFT) with the PBE functional³⁶ using a DKH-def2-SVP basis set³⁷ including the relativistic effects via the 2nd order Douglas-Kroll-Hess formalism.^{38–40} The calculations were performed with the Orca 4 quantum chemistry package.⁴¹ Simulations of the Mn K-edge XANES spectra were performed using the FDMNES package.^{42,43} Throughout a self-consistent muffin-tin-type potential of radius 6.0 Å around the absorbing site was employed. The interaction with the X-ray field was described using the electric quadrupole approximation, and scalar relativistic effects were included. To transform the computed cross-sections into XANES spectra that can be compared to experiment, the cross-sections were convoluted with a function that accounts for the core-hole-

lifetime broadening, instrument response, and many-body effects, e.g., inelastic losses. Throughout this work, this convolution has been performed using an energy-dependent arctangent function⁴⁴ via an empirical model close to the Seah-Dench formalism.⁴⁵

XAS were computed as a function of displacements along dimensionless normal coordinates (DNC). In practice, we have generated a set of geometries that each correspond to a displacement along a given DNC. The displacement along mode 60 was chosen by simple comparison of the calculated XAS of distorted geometries to the experimental ones. Because experimentally, not all molecules are excited, the resulting XAS will include both molecules in the excited states and in the ground states. In order to take into account this effect in the calculation, the ratio of the ground state to excited state signal must be simulated, which will modulate the amplitude of the signal.

* system *

4. It is not clear how the specific system could be used in a all-optical device. How would you read and write ? This is connected to the point 1. In normal magnet a stronger than coercive field is used to change the magnetization...

ANSWER: Reading and writing could be done optically. There is a whole field in condensed matter physics dedicated to using femtosecond laser pulses to read and write “normal” magnetic materials, with the aim to create ultrafast all-optical devices. State-of-the-art in this field is to switch the magnetic anisotropy, and thereby reverse the magnetisation direction completely, by exciting d-d transitions in a cobalt-doped yttrium iron garnet (Co:YIG) [Stupakiewicz *et al.*, *Nature*, **542**, 71 (2017)]. This provides a method to write to memory (for example to overwrite a “1” to a “0”). There would therefore not be another “normal magnet” with a stronger coercive field, but the anisotropy of the material is modulated so that the magnetisation changes direction. By choosing the pump polarisation, the direction of the anisotropy change can be controlled, and the final magnetisation direction is in turn controlled (see Stupakiewicz *et al*). The read-out is envisaged to be an optical probe that measures the magnetic state using Faraday rotation (magnetisation-sensitive probe affecting the polarisation state of the light beam). We have added to the section in the introduction describing this method:

Presently, magnetic data storage in servers use a small electromagnet to switch the magnetisation direction when overwriting stored data from, say, ‘0’ to ‘1’. Despite impressive developments associated with this method,¹ read-write speeds cannot be faster than nanoseconds.² In the last couple of decades there has been a strong research focus on photomagnetic data recording, where femtosecond laser pulses, **instead of external magnetic fields**, are used to reverse the magnetisation direction within tens of picoseconds.³ State-of-the-art methods in this area involve using either electronic^{4–6} or phonon^{7,8} excitation to control the magneto-crystalline anisotropy via changes in the crystal environment upon excitation. This can lead to a torque that switches the magnetisation direction into a different orientation, which could lead to a new, faster way of storing data. **The readout in this case is done using the Faraday effect.**^{4–8}

In single-molecule magnets, the magnetic bidirectionality comes from the magnetic anisotropy (as mentioned above in the reply to Point 1). We envisage a similar process to that in Co:YIG, whereby we excite d-d transitions that will change the magnetic anisotropy and force the magnetisation to change direction. This would correspond to writing data, and reading could be achieved using Faraday rotation. Both read/write processes can be focused to the single-molecule level using plasmonics (we mentioned this in the introduction: “SMMs have the added advantage of operating on the molecular scale, which could significantly increase data storage density when combined with optical techniques and plasmonic technologies to address individual molecules.¹⁰”). We fully acknowledge that this might take 10 years to develop and that there are many issues, but the results on Co:YIG are very promising in this direction.

Once changes the magnetization is stable for very long time. What about the Mn3 system ?

ANSWER: As mentioned in the response to Point 1, the zero-field splitting magnetic anisotropy does lead to a barrier for magnetisation reversal. For Mn(III) SMMs, this is quite low, so cryogenic temperatures are needed to retain the magnetisation direction (which should be stable for weeks). New developments in molecular magnetism are happening quickly and there are now several single-molecule magnets that show hysteresis above liquid nitrogen temperatures, so SMMs will be able to retain the magnetisation for long periods of time. We stress again that ultrafast dynamics of SMM is in its infancy and new methods (like the ones presented here) are needed to probe them; to develop new methods you need a good model system, which the Mn(III)-based SMMs are.

* minor point *

more specific information on the experimental setup should be provided, flux per X-ray pulse, kinds of detectors used. One can only guess that the signal was measured in total fluorescence yield ...

ANSWER: We thank the reviewer for this comment and have added more details in the methods section. We did indeed write in the first submission that we had measured the total fluorescence yield but have made this clearer now:

The TR-XAS measurements were performed at BL3³³ of SACLA.³⁴ The X-ray absorption was measured using the total fluorescence detection method. For both incident and total fluorescence X-ray intensity measurements, we used Si photodiodes (Hamamatsu, S3590–09). The geometry of these detectors is described in Ref. [Struct. Dyn. 6, 054302 (2019); doi: 10.1063/1.5111795]. Two Si(111) crystal/s in the (+,-,-,+) geometry were used to monochromate the beam and a Be compound refractive lens was used to focus the X-rays at the sample position to a size of 10 (H) x 10 (V) μm^2 at FWHM. At the interaction point, 400 nm laser pulses were focussed down to 210 (H) x 215 (V) μm^2 at FWHM and interleaved with the X-ray pulses to measure both optical laser on and off spectra to acquire the pump induced difference spectra. The timing jitter was measured using the beamline diagnostics³⁵ and corrected in the data analysis, to improve the time resolution. We estimate that X-ray flux per pulse was ca. 4 μJ . This estimation is based on the bandwidth of the Si double crystal monochromator (which only preserves ca. 2% of the SASE pink beam energy of 600 μJ) and that transmission through the Be compound refractive lens is 40% at 6.5 keV, leaving ca 4 μJ /pulse.

All is all I feel that this paper tries to struck a balance between the new opportunities in time-resolved XAS and the interest in molecular magnet. In practice though, I feel that both aspect are not quite compelling enough for publication in Nature Communication. The X-ray data are nice but they are not striking enough to be a landmark in the field, the system itself, despite of obvious interest for photo-physicists, is not quite the best magnet due to the short lifetime at the very least.

ANSWER: The field of ultrafast photoinduced dynamics in single-molecule magnets is still in its infancy. There are a lot of open questions, such as which are the best methods to study these large and complicated molecules, and what are the best molecular properties one should look for. Furthermore, once some initial insights are gained, then what are the best ways to measure and control the magnetisation in the excited state. Therefore, pushing the boundaries for new measurement techniques is imperative for driving the field forward. We really do believe that this is a landmark achievement in the field of photophysics of molecular magnets because we have shown on a model system that it is possible to measure the extent of the change of Jahn-Teller distortion, which in turn is paramount for controlling the magnetisation in single-molecule magnets. This, and in combination with the large range of chemistry and physics communities interested in our work, we believe makes Nature Communications the best suited journal for our work.

Reviewer #2 (Remarks to the Author):

The authors have improved the manuscript and clarified some important aspect.
I thus recommend publication without further review needed from my side